# Selection and Comparative Gene Expression of Midgut-Specific Targets for *Drosophila suzukii*

**DOI:** 10.3390/insects14010076

**Published:** 2023-01-12

**Authors:** June-Sun Yoon, Seung-Joon Ahn, Man-Yeon Choi

**Affiliations:** 1USDA Agricultural Research Service, Horticultural Crops Research Unit, Corvallis, OR 97331, USA; 2Department of Agricultural Convergence Technology, Jeonbuk National University, Jeonju 54596, Republic of Korea; 3Department of Biochemistry, Molecular Biology, Entomology and Plant Pathology, Mississippi State University, Mississippi State, MS 39762, USA

**Keywords:** midgut transcriptome, differential gene expression, *Drosophila suzukii*, spotted-wing drosophila, GPCRs

## Abstract

**Simple Summary:**

Spotted-wing drosophila (SWD), *Drosophila suzukii*, a major pest of small fruits and cherries, is often managed with conventional insecticides. RNA interference (RNAi) technology has been investigated to develop an alternative control method for SWD. Our previous RNAi work showed that the RNAi efficacy on SWD was limited because double-stranded RNA (dsRNA) was degraded by the midgut nuclease (= dsRNA − degrading enzyme) before passing through the midgut membrane of the fly. To overcome this obstacle, RNAi directly targets the midgut genes, thus eliminating the need for dsRNA to pass through the midgut membrane. The primary focus of this study is to identify the fly adult midgut’s genes. We compared differential gene expressions between the midgut and the rest of the whole-body using transcriptomic analysis. We found that 1921 genes were upregulated, and 1834 genes were downregulated in the midgut. We chose ten midgut-specifically upregulated genes involved in various biological functions and confirmed their gene expressions. Particularly, the midgut membrane proteins found in this study would be potential targets for developing RNAi and other biological tools for controlling SWD in the future.

**Abstract:**

Spotted-wing drosophila (SWD), *Drosophila suzukii*, is a destructive and invasive pest that attacks most small fruits and cherries. The current management for SWD involves the use of conventional insecticides. In an effort to develop a biologically based control option, the application of RNA interference (RNAi) has been investigated. To develop an RNAi approach, suitable targets must be identified, and an efficient delivery method must be developed for introducing the double-stranded RNA (dsRNA) in the midgut. In *D. suzukii*, we previously found that dsRNA nucleases actively degrade dsRNA molecules in the midgut. In this study, we focused on identifying biological targets focused on the midgut membrane. The profile of midgut-specific genes was analyzed and compared with the genes expressed in the whole-body using transcriptome analysis. Differential gene expression analysis revealed that 1921 contigs were upregulated and 1834 contigs were downregulated in the midgut when compared to genes from other body tissues. We chose ten midgut-specifically upregulated genes and empirically confirmed their expressions. We are particularly interested in the midgut membrane proteins, including G protein-coupled receptors (GPCRs) such as diuretic hormone 31 (DH31) receptor, neuropeptide F (NPF) recepror, toll-9, adhesion receptors, methuselah (mth), and gustatory receptor, because insect GPCRs have been offered great potential for next-generation pest management.

## 1. Introduction

Spotted-wing drosophila (SWD), *Drosophila suzukii*, is a worldwide pest fly. Females have a serrated ovipositor, a morphological adaptation [1] that enables them to lay eggs into ripening fruits before harvest. Moreover, *D. suzukii* exhibits a wide host range in its introduced and native range, infesting most small fruits and cherries in Asia, Europe, and North America [2], and now they have spread to subtropical areas including South America, Australia, and Africa [3,4]. Therefore, small fruit industries around the world are facing economic losses due to increased spending on pest management in the field.

Despite many potential adverse effects, current control methods for *D. suzukii* rely primarily on conventional chemical insecticides. Recent research has sought to develop alternative control methods to replace or at least reduce chemical insecticide inputs [5,6], but they cannot fully protect small fruits from fly damage. The availability of various insect genomes and RNA sequences facilitates a genome-based approach to identifying biological targets for insect control. Since the genome of *D. suzukii* was published [7], a few case studies analyzing the functional transcriptome of its antennae were reported [8,9,10]. A comprehensive genome study was recently achieved by the long-read sequencing [11]. Moreover, related to pest management, transcriptome studies facilitate finding target genes for RNAi and can be applied towards developing sterile insect technique (SIT) and CRISPR/Cas9 technology reviewed by [12].

Insect neuropeptides (NPs) and G protein-coupled receptors (GPCRs) are involved in a variety of critical physiological processes such as homeostasis, feeding, digestion, excretion, circulation, reproduction, and metamorphosis during insect life stages [13,14]. Therefore, insect NPs and their GPCRs have been proposed as potential targets for decades [15,16,17]. Recently, the application of RNA interference (RNAi) has been investigated to develop a new biological alternative, a more species- and gene-specific method for controlling pests. A few trials for *D. suzukii* RNAi with potential targets evaluated the RNAi effects against larvae and adults [18,19,20,21]. However, the RNAi efficacy with the target genes was limited because the dsRNA molecules introduced in the midgut are degraded before being transported into hemocoel [22]. There are some options to overcome this obstacle: (1) encapsulation of dsRNAs using nanoparticles or exosomes, and (2) direct-targeting the midgut membrane for RNAi, thus eliminating the need for dsRNA to pass through the midgut membrane, and (3) modification of dsRNA structures to avoid/reduce the degradation [23,24]. 

The digestive tract, especially the midgut, is an important alimentary canal where various mechanical and physiological processes occur, including food digestion, nutrient absorption, and detoxification [25]. The importance of the gut is evident by the many studies of gene expression profiles of the gut using various advanced technologies during the past decades. Moreover, studies of microbes, insecticide resistance, immunity, and nutrient absorption have been accelerated to understand gut physiology and its application [26,27].

Regardless of extensive gene sequencing data, basic research on the gene expression of the midgut and the rest of body and differential data comparisons in *D. suzukii* is limited. Therefore, the primary focus of this study is to identify the adult midgut membrane genes for potential targets with particular interest in membrane-bound proteins, including GPCRs. Differential gene expressions of the midgut and whole-body of *D. suzukii* were analyzed using transcriptome. The selected genes showing relatively high expression in the midgut were validated by qRT-PCR. In addition, gene ontology analysis was performed to identify the midgut-specific genes involved in various biological functions and differential expression patterns. This midgut transcriptome data might be useful to those who work on gut physiology, gut microbes, insecticide resistance, and the selection of biological targets to develop *D. suzukii* control methods. 

## 2. Materials and Methods

### 2.1. Insects

*Drosophila suzukii* were maintained using standard rearing methods [28] at the Horticultural Crops Research Unit, USDA ARS in Corvallis, OR, USA, and 3 to 5-day-old adults were used in all the experiments in this study. 

### 2.2. Isolation of Total RNA 

Approximately 20–30 midguts were dissected from 3 to 5-day-old adults and collected in a 1.5 mL tube on dry ice. Samples were homogenized in a lysis buffer using a plastic pestle. Total RNA was isolated using PureLink RNA Kit (Thermo Fisher Scientific, Waltham, MA, USA) according to the manufacturer’s instructions. The total RNA was quantified on a NanoDrop 2000 spectrophotometer (Thermo Fisher Scientific) and stored at −20 °C until use. For comparison, total RNA was isolated from the whole-body of female adults using the same kit above. The total RNA isolated from the female midgut and whole-body were used for transcriptome sequencing and differential gene expression analysis. The RNA concentrations obtained after the extraction were adjusted to 60 ng/μL in 50 μL for both tissues. The quality of RNA was analyzed using Agilent 2100 Bioanalyzer (Agilent Technologies, Santa Clara, CA, USA) based on RNA Integrity Number (RIN). 

### 2.3. Illumina Sequencing

The cDNA libraries were prepared using TruSeq Stranded Total RNA Library Prep Kit (Illumina, San Diego, CA, USA). Illumina sequencing was performed by Psomagen, Inc. (Rockville, MD, USA) using the Illumina HiSeq 2000 platform. Briefly, ribosomal RNA was removed from total RNA, and the remaining RNA was purified, fragmented, and primed for cDNA synthesis. The first strand of cDNA was synthesized with reverse transcriptase and priming with random hexamers. The RNA template of the first strand cDNA was replaced by incorporating dUTP in place of dTTP to generate the second strand cDNA. The cDNA then underwent an end repair process, adenylation of 3’ ends, and subsequent ligation of the adapter. The quantified and qualified libraries for both samples were sequenced using an Illumina HiSeq 2000. RNA-Seq raw reads are available in the GenBank SRA database (BioProject accession No PRJNA903844).

### 2.4. Anlaysis of up- or down-Regulated Genes

Based on the expression values as fragment per kilobase of transcript per million mapped (FPKM) reads up- or down-regulated genes were identified among the midgut and whole-body samples (MG and WB) after transformation, normalization, and fold change (fc) comparisons. The FPKM values were obtained by the Reference Annotation Based Transcript (RABT) method using -G option of Cufflinks. Raw FPKM values were transformed into Log_2_(FPKM+1) values in order to make the range with low values widely distributed, and then normalized by quantile normalization method using preprocessCore in R package in order to reduce systematic bias among the samples. Up- or down-regulated genes with |fc| ≥ 2 were regarded as up- or down-regulated genes. If the FPKM value was 0 (i.e., not changed) in either of the two samples, it was not included in the analysis. However, they were retrieved from the original raw data and manually added to the DEG analysis to identify tissue-specific genes.

### 2.5. Functional Annotation and Gene Ontology Analysis

For the two tissue samples, if more than one FPKM value was 0, it was not included in the analysis. Therefore, from the total of 13,583 contigs, 4712 were excluded, and only 8871 contigs were used for statistical analysis. The functional annotation of the 13,583 official gene set (OGS) was performed by sequence similarity searches against the NCBI non-redundant protein sequences (nr) database using BLASTx algorithm with a cut-off E-value of 1 × 10^−5^ run by Geneious 8.1.5 software (Biomatters Ltd., Auckland, New Zealand). In cases where there were no clear “hits,” the sequences were searched again using BLASTn. The BLAST hits were grouped into major gene families based on their putative functions in fly biology. The organism information obtained by the BLAST hit was collected to confirm the sequence similarity of the transcripts to closely related species. Gene ontology analysis was performed via ShinyGO 0.76 (http://bioinformatics.sdstate.edu/go/, accessed on 15 November 2022). Biological processes, cellular components, and molecular functions are examined. 

### 2.6. Manual Annotation of the Midgut Transcriptome

Regarding 1921 upregulated genes, we searched for the DS10 numbers on SpottedWingFlyBase (http://spottedwingflybase.org/, accessed on 15 November 2022). Then, the C.G. number, FBgn, and symbol were found in the *D. melanogaster* FlyBase (https://flybase.org/, accessed on 15 November 2022). Functional analysis and InterPro information were also noted in Appendix A. 

### 2.7. Quantitative Real-Time PCR (qRT-PCR)

To validate midgut-specifically upregulated genes from the fly pool, we selected some putatively important midgut genes, including dsRNases, GPCRs, and neuropeptides, and verified their expression patterns by qRT-PCR using samples isolated from different sets of flies. To isolate total RNAs, three different replications of the midgut pool from 3 to 5-day-old adults were dissected as described above. cDNAs were synthesized from 500 ng total RNA for the midgut and whole-body samples, respectively, using Verso cDNA Synthesis Kit (Thermo Fisher Scientific), according to the manufacturer’s instructions. The synthesized cDNAs were stored at −20 °C until qRT-PCR. Gene-specific primers (Appendix A) were designed to amplify 90–120 bp portions of target genes and two reference genes. The qRT-PCR reaction mixture was prepared in 20 μL with 2X PowerUp SYBR Green Master Mix (Thermo Fisher Scientific), deionized H_2_O, and 0.25 μM primer pairs. The reaction conditions were 95 °C for 15 min and then 40 cycles of 95 °C for 15 s and 60 °C for 1 min, run in StepOnePlus™ Real-Time PCR System (Thermo Fisher Scientific). This was followed by a melting curve analysis over the range of 60–95 °C and sequencing of the qRT-PCR products to check the amplification specificity. A minimum of three repetitions of each standard were done using different preparations of *D. suzukii* cDNA. Five different housekeeping genes were tested for primer efficiency, and *Rpn2* and *Rpt6* were selected as reference genes in the analysis because they were less variable than other housekeeping genes previously tested [9]. The geometrical mean (geomean) of *Rpn2* and *Rpt6* was used for the qRT-PCR experiment, and the delta delta Ct method was applied.

### 2.8. Statistical Analysis

Student’s *t*-test was used to compare the relative mRNA level differences between control and treatment groups to test the relative mRNA expression. A *p*-value of 0.05 or less between groups was considered significantly different.

## 3. Results and Discussion

### 3.1. Midgut Transcriptome Analysis

Transcriptomes of midgut and whole-body generated over 5.6 G bases, resulting in 54 and 59 million reads, respectively (Appendix A). Based on the *D. suzukii* official gene set (OGS) provided in the SpottedWingFlyBase (http://spottedwingflybase.org, accessed on 15 November 2022), over 78.9% and 78.8% of reads from the midgut and whole-body transcriptomes, respectively, were mapped to the reference genome and assembled into 13,583 contigs (Appendix A).

For expression analysis, contigs with zero value of FPKM in either transcriptome were excluded from analysis, leaving 8871 contigs to be further analyzed. The distributions of the contig expression levels were similar between the midgut and whole-body (Figure 1).

The analysis of up- or down-regulated genes between the two transcriptomes revealed 1921 contigs that were upregulated and 1834 contigs that were downregulated in the midgut compared to the whole-body (Figure 1), whereas the rest (6881 contigs) were similarly expressed between the two samples. To provide a functional overview of the transcriptional differences between midgut and whole-body, gene ontology (G.O.) enrichment analysis was performed (Figure 2). According to the ontology data, signaling-related genes are dominantly occupied in the biological process, and membrane-related genes (plasma membrane and endomembrane system) are highly present in cellular components. In molecular function analysis, hydrolase activity, cation, and metal ion binding take large portions.

The most upregulated genes of the midgut compared to whole-body are listed in Appendix A. Sequences of DS10 contigs were searched against FlyBase (version FB2022_05; http://flybase.org/, accessed on 15 November 2022) to find homologous genes in *D. melanogaster* using the BLASTn program with the BLOSUM62 matrix at the threshold value 0.01. Hence, corelated CG numbers, molecular functions, and biological functions were noted (Appendix A). Moreover, InterProScan data was added to classify the protein family of each gene. Using the Drosophila Gene Expression Tool (DGET), a search program of RNA-Seq-based expression profiles (www.flyrnai.org/tools/dget/web/, accessed on 15 November 2022), all the genes were checked. Most of the genes were highly expressed in the digestive system [29] (Appendix A). These genes showed a relatively higher expression pattern in the digestive system than in other body parts.

### 3.2. Validation of the Midgut-Specifically Upregulated Genes

In order to validate FPKM values of transcriptome data, ten genes were selected from the upregulated gene group, confirmed by qRT-PCR, and compared to transcriptome data (Figure 3 and Figure 4). Besides the qRT-PCR confirmation, to overcome the limit of experimental replications and to avoid false-positive-effects, the upregulated genes from the midgut of *D. suzukii* were compared with *D. melanogaster*’s expression database (Appendix A). Those ten genes were selected by previous literature to prove the midgut highly expressed characteristics described below. The midgut is the primary digestive system in insects for the uptake of biological molecules, including nutrients and bioactive compounds (i.e., dsRNA and small peptides) transportation into the target tissues or cells through the hemocoel. From the midgut transcriptome analysis, we focused on identifying potential biological targets to be applied for RNA interference (RNAi) or receptor interference (Receptor-i) for *D. suzukii* management. RNAi for dipteran pests is challenging with low efficiency of RNAi due to limited dsRNA delivery and degradation of dsRNA by the midgut nuclease (=dsRNase) prior to transport into target cells through the midgut [19,22,30]. 

Considering other insects, it is common knowledge that expression patterns of dsRNases are relatively high in midgut compared to other body tissues [31,32,33,34,35]. In our previous study, two *D. suzukii* dsRNase genes, DrosudsRNase1 (DS10_00008824, GenBank Accession No. MW984608) and DrosudsRNase2 (DS10_00008823, GenBank Accession No. MW984609), were identified, functionally expressed in Sf9 cells derived from *Spodoptera frugiperda*, and characterized for enzyme activity [22]. Both dsRNA degradation enzymes are active in the midgut; dsRNase1 is relatively more active than dsRNase2 in terms of its gene expression level and in vitro degradation activities [22]. From the RNA-Seq analysis, the FPKM expressions of the dsRNase1 & 2 genes in the midgut were at similar levels, which are 3.7 and 3.9 times greater than whole-body, respectively (Appendix A). In the qRT-PCR result, their relative mRNA levels (=FC values) of midgut tissues are 35.6–52.2 times greater than whole-body. The previous and current qRT-PCR results confirmed that dsRNase1 is a dominant enzyme in *D. suzukii*.

### 3.3. Midgut-Specifically Upregulated Membrane Proteins and GPCRs

Except for the two dsRNA enzymes above, the other eight membrane proteins selected in this study have not yet been identified and characterized in *D. suzukii*. DS10_00008095 is orthologous to the *D. melanogaster* CG6981 (Snakeskin), which plays an important role in the gut: intestinal epithelial cell development, the establishment of endothelial barrier, anterior midgut development, and septate junction assembly. Expression of Snakeskin in the midgut is significantly high compared to the whole-body, which ranges 13.0–19.5 times from RNA-Seq and qRT-PCR results of *D. suzukii*. As a pest management tactic, dsRNA targeting the Snakeskin gene of corn rootworm induced larval growth inhibition and eventual mortality [36]. Midgut-specific genes can be targeted for RNAi-mediated pest control because they can cause an instant effect, such as fasting and mortality, by interfering with the midgut, where digestive substances are absorbed.

Over decades, insect GPCRs have been proposed as potential targets for next-generation pest management because they involve various critical physiological processes at all insect life stages [15,37,38]. GPCRs are membrane-bound proteins consisting of seven transmembrane domains. Especially, the GPCRs upregulated in the midgut mediate various endocrine signals including neurohormones into cells for digestive processes, activation of digestive enzymes and gustatory receptors, recognition of nutritional levels, and water balance [16,38]. These could be potential targets for *D. suzukii* control. The midgut transcriptome profile selected the seven most upregulated GPCRs in the midgut (Figure 5), as described in detail below.

DS10_00007171 (GenBank Accession No. XP_036670585) is a toll-like receptor. Toll receptors, in general, play a crucial role in the innate immune system and interact with microbes. Moreover, epithelial toll-like receptors are specialized in gut homeostasis, nutrition absorption, and gut-related diseases [39]. Expression levels of the toll receptor (=Toll-9) from the RNA-Seq and qRT-PCR results in the midgut are 19.1–20.3 times greater than the whole-body (Figure 4), indicating Toll-9 might be a midgut-specific gene in *D. suzukii*. The biological function(s) of this gene in the midgut remains to identify in the future study. In insects, the *Bombyx mori* Toll-9 gene is engaged in the local gut immune response [40]. Of 14 putative *B. mori* toll genes, *B. mori* Toll9-1 is upregulated explicitly in the midgut compared to other body parts [40].

DS10_00006592 is orthologous to CG17415 of *D. melanogaster*, which is related to a member of the calcitonin-like receptor (CLR) and identified as the diuretic hormone 31 receptor (DH31-R) [41]. DH receptors belong to the Class B GPCR group signaling pathway using cAMP as a second messenger instead of Ca^2+^ [16,42]. In the qRT-PCR result, the expression of DH31-R showed midgut is almost 40 times (fold change ratio of MG/WB) greater than whole-body, indicating DH31-R might be involved in the digestion of *D. suzukii*. The physiological function of the DH31 receptor is expected to regulate fluid secretion from foods in the midgut through Malpighian tubules in Drosophila [41]. DH31-R would be a good target because water regulation is critical in dipteran insects. The other diuretic hormone receptor in insects is the DH44 receptor (DS10_00002631). Its expression in the *D. suzukii* midgut was as high as that of DH31-R from the transcriptome analysis.

DS10_00013361 gene (GenBank Accession No. XP_036670319) refers to methuselah (mth) that belongs to the Class B GPCR group, related to increasing the lifespan with increased resistance to heat and oxidative stress, and starvation in Drosophila mutants [43,44]. The gene has been evaluated to understand its potential functions, including its response to insecticide stresses and its role in the regulation of longevity [44], pointing out a potential target for new management of *D. suzukii*. The gene expressions of the midgut measured by RNA-Seq and qRT-PCR are 8.8–7.5 times greater than whole-body, which is significant.

Both DS10_00012673 and DS10_00012139 (GenBank Accession No. XP_016939533 and XP_016935592) were identified for adhesion GPCRs, one of the glycoproteins, E-cadherin, that functions as part of a dynamic membrane-spanning macromolecular complex to facilitate adhesion mechanically between epithelial cells of the midgut [45,46]. In addition, the receptor extends to multiple functions, including cell recognition, sorting, boundary formation, maintenance, and coordination of cell movements. There are different adhesion receptors; both receptors belong to Ca^2+^-dependent Type 1 classical cadherins, like housekeeping genes, which are expressed in almost all tissues [45]. In *D. suzukii* midgut, two adhesion GPCRs are found, that adhesion1 receptor (Adsn1-R) is dominant and expressed at 17.7–25.5 times greater than whole-body from RNA-Seq and qRT-PCR analyses, respectively. The expression of Adsn1-R is ~5 times greater than Adsn2-R in the midgut, which would be a better target for the fly. However, the specific functions of the two GPCRs are unclear in *Drosophila*.

DS10_00008044 (GenBank Accession No. XP_036675103) was identified with the receptor of neuropeptide F (NPF), which has various functions including food search, intake, and odor perception important in the regulation of feeding activities in insects [13,14]. In *Drosophila*, the first NPF receptor (NPF-R) exhibits a similar pattern of localization in larvae, as the NPF ligand gene that is expressed in the brain and in the midgut for regulation of feeding and digestion [47]. As expected, the NPF-R was highly expressed in the midgut of *D. suzukii*, 3.0–4.4 times greater than whole-body from both analyses, suggesting its role in modulating food intake and ingestion with other neuropeptides in the midgut. Interfering with the specific receptor would be critical to the physiological function of the NPF in the midgut.

DS10_00005654 (GenBank Accession No. XP_016941187) was identified as a gustatory receptor 43a (Gr43a), one of the most conserved gustatory receptors across insect species. The specific gustatory receptor is highly expressed in the midgut, 2.6–4.8 times greater than whole-body in *D suzukii*. The receptor is usually expressed in the brain and plays a critical role in sensing internal fructose levels in Drosophila [48,49]. Activation of this receptor is mediated by the neuropeptide corazonin, as a neurotransmitter, in the fly [49]. The Gr43a neurons projecting to the midgut and expression of Gr43a in the gastrointestinal system are found across different insect groups that may regulate food transport and/or secretion of neuropeptides/hormones in response to sugar consumption [49]. Like the NPF-R, disruption of the Gr43a pathway would be critical to the sugar metabolism and/or nutritional value of food for the fly survival.

All the upregulated genes for biological targets were provided in this study (See Appendix A). Due to the contig-level assembly process and deficient *D. suzukii* annotation information, duplicated annotations exist, and many unknown genes are listed. Although the physiological processes of the highly expressed genes selected in this study are inferred from other species, further studies are needed to verify their specific functions in *D. suzukii*. While this RNA-Seq data may not give the exact values for gene expression levels, it not only provides overall gene expression patterns for the study of midgut physiology, but also for gut microbes, insecticide resistance, and identifying biological targets for *D. suzukii* management. 

## Figures and Tables

**Figure 1 insects-14-00076-f001:**
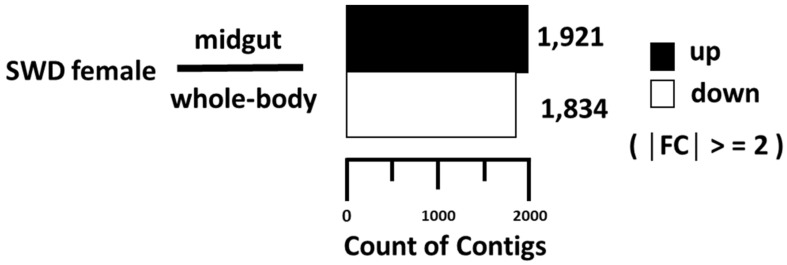
Up- and down-regulated gene counts in *Drosophila suzukii* female midgut compared to its whole-body. 1921genes were up-regulated, and 1834 genes were down-regulated.

**Figure 2 insects-14-00076-f002:**
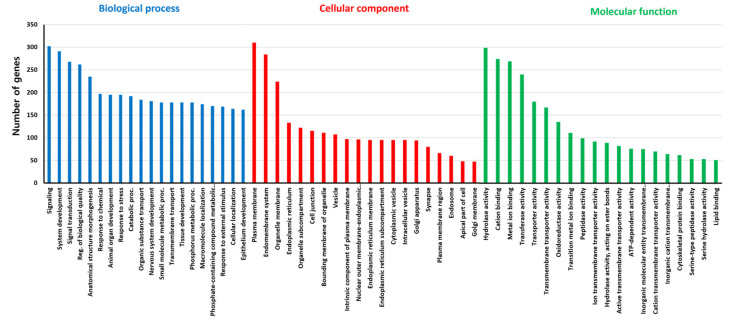
Gene ontology terms of *Drosophila suzukii* midgut-specific up-regulated genes for the three main biological categories.

**Figure 3 insects-14-00076-f003:**
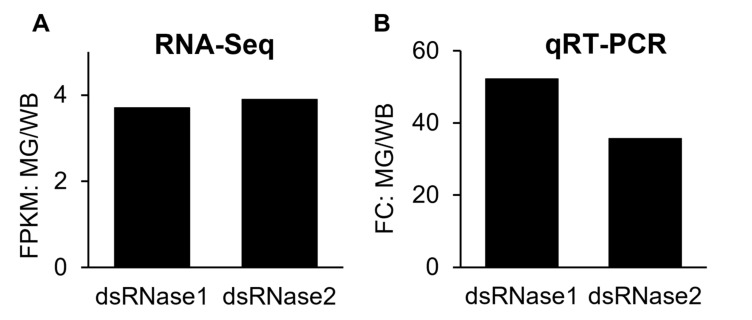
Expression of dsRNase genes in the midgut of *Drosophila suzukii* as measured by RNA-Seq (**A**) and qRT-PCR (**B**). Ratios of FPKM (fragment per kilobase of transcript per million mapped reads) and FC (fold change) of the midgut (MG)/whole-body (WB).

**Figure 4 insects-14-00076-f004:**
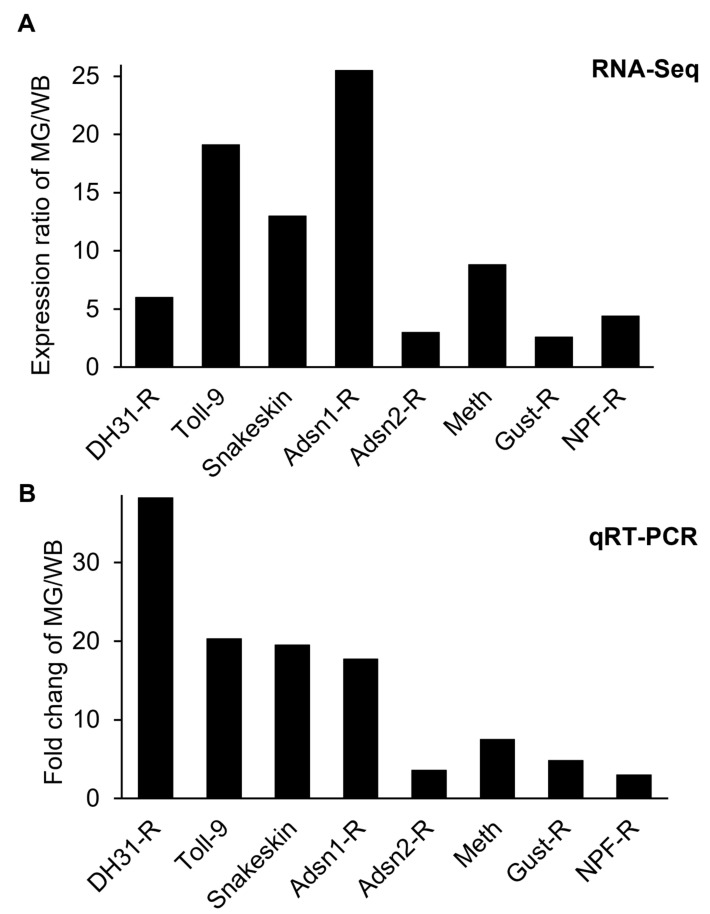
Expression of membrane protein genes in the midgut of *Drosophila suzukii* as measured by RNA-Seq (**A**) and qRT-PCR (**B**). Ratios of FPKM (fragment per kilobase of transcript per million mapped reads) and FC (fold change) of the midgut (MG)/whole-body (WB). DH31-R: diuretic hormone 31 receptor; Adsn1&2-R: adhesion receptor1 and 2; Meth: methuselah; Gust-R: gustatory receptor; NPF-R: neuropeptide F receptor.

**Figure 5 insects-14-00076-f005:**
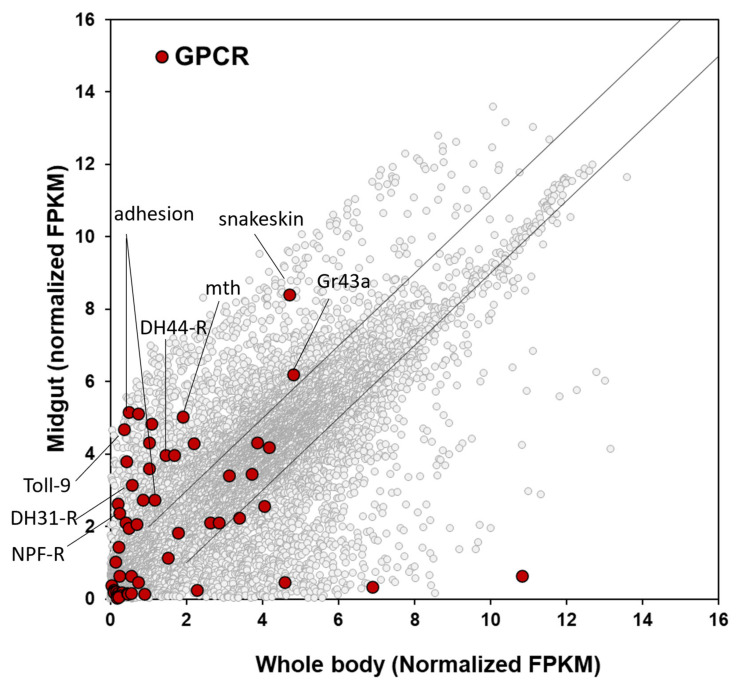
Scatter plot of the normalized gene expression levels of GPCRs between midgut and whole-body. Gene expression levels of different genes families are highlighted in different colors on the gray background scatter plot of total genes (8871) expressed in both midgut and whole-body. Two diagonal lines in the plot refer to │fold change (fc)│ = 2. Snakeskin is a midgut membrane protein, not a GPCR.

## Data Availability

The data presented in this study are available in [insert article or Appendix A here].

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
