# Peer review of "Selection and Comparative Gene Expression of Midgut-Specific Targets for Drosophila suzukii"

_insects, 2023, doi:10.3390/insects14010076_

Round 1

Reviewer 1 Report

The current article by Yoon et al. describes the selection and comparative gene expression of midgut-specific targets that might become potential targets for developing RNAi and other biological tools for controlling Drosophila suzukii in the future. For this purpose, they used transcriptome data from adults' midgut tissue and successfully found several relevant target genes. This study is interesting and might be useful as a piece of fundamental information to those researchers working to develop biological tools for controlling D. suzukii.

However, the description of some very important points was inadequate or completely missing. I have little confidence in the methodology section and came away with too many questions; thus, I would like to recommend a major revision before the publication of this manuscript.

Line 72-75: Are the authors quite sure that there are only two methods? Could you please provide a reference?

Line 101-105 or the whole paragraph: There is no information regarding the number of replicates. The authors must provide this information. Moreover, do 20-30 midguts represent one replicate is not clear?

Line 107-108: I am wondering whether female adults were selected for midgut samples for comparison female adults were selected? If not, then why were different sexes selected for comparison? Moreover, were the female adult sample from 3-5-day old?

Line 109-111: Please describe the concentration average of Total RNA obtained after the extraction and the quantity of total RNA used for further experiments.

Line 121-122: How many libraries were generated and sequenced is unclear.

Line 136: To follow more stringent criteria, I would like the authors to remove those contigs having FPKM lower than 10 or at least 5 in both samples.

Line 195-196:  Please express the number of regulated genes in percentage to compare the transcriptional changes more clearly. Moreover, compare the transcriptional changes in D. suzukii midgut with other research. Did regulated genes show the same pattern (up or down)? Explain whether the level of down-regulation was greater or up-regulation.

Line 208-209: Please provide more information for this part. Rewrite the Gene Ontology results according to the percentage of the genes involved in the functional grouping.

Line 228: I would like to suggest that this statement should be after line 232.

Line 247: I would like to suggest writing the technical name of the pest here.

Unfortunately, I lost interest in the last section after line 242, as there is not much interesting information to attract the readers. There are mere explanations of other studies, and it seems like a literature review rather than a discussion. I would like to suggest that this part should be improved.

Here are a few suggestions:

1.     Please describe how much up-regulated midgut genes of the current study were (in folds) compared to similar studies.

2.     Mention the differences in expression pattern or level, and then add logical reasons why there might be differences.

3.     Explanations should also include and be based on functions of these genes, not only mere expression. 

Reviewer 2 Report

In this manuscript, Yoon et al. uses RNA-seq technology to identify potential gene targets expressed in D. Suzukii midgut for population control. The objectives of this study are clear and the manuscript is well written. Though, due to statistics of low power and lack of additional analyses, the specificity of the proposed targets is not fully demonstrated.

Major

* The differential genes are derived from the comparison of only 2 replicates each from a different biological sample (1 replicate vs 1 replicate). Even if each replicate is a pool of material from several insects, this comparison is likely to produce results full of false positives. A minimum of 3 replicates per sample would have allowed to control this problem.

* From the RNA-seq results, the authors decide to validate the expression of 10 genes by RT-PCR. As explained by the authors, those genes are already known for their expression in the gut. Were they all (each) known for D. Suzukii? This should be made clear in the text.

* Authors propose in particular 10 midgut specific genes as target for gene silencing for population control. Is it possible to build RNAi system to target those genes specifically in D. Suzukii and not in other insects or living organism? This should be investigated for example by comparing the RNA sequencing results to existing sequence databases.

* Discuss better the limits of the study, especially the uncertain relevance of the long list of gene targets

Minor

* L18: deferential => differential

* L130: quintile => quantile

* L203: „blasted“ is not an English verb, please rephrase and include version number and parameters

* Table S3: what are the numbers in the table ? Please describe bertter the table and explicitely the numbers unit.

Round 2

Reviewer 1 Report

A minimum of three replicates is required for a robust study as less than three replicates might produce false positive results, and I believe this is the major issue with this study that should be resolved. Although authors have used three replicates in qPCR, that is not enough to claim that the results are robust, and authors must consider this issue. 

Author Response

We appreciate your comments/suggestions and the reference paper. We agreed and replaced DEG with ‘up- or down-regulated genes’ for the title of Section 2.4 and the other parts. We also revised with midgut-specifically upregulated genes in Sections 3.2 and 3.3.  

Round 3

Reviewer 1 Report

The authors have not answered my comments on round 2. Moreover, I did not suggest any reference paper to follow. Here are my comments on round 2 and now for round 3 also. 

A minimum of three replicates is required for a robust study as less than three replicates might produce false positive results, and I believe this is the major issue with this study that should be resolved. Although authors have used three replicates in qPCR, that is not enough to claim that the results are robust, and authors must consider this issue. 

Author Response

Responses to the reviewer's comments

"A minimum of three replicates is required for a robust study as less than three replicates might produce false positive results, and I believe this is the major issue with this study that should be resolved. Although authors have used three replicates in qPCR, that is not enough to claim that the results are robust, and authors must consider this issue. ".
I agree with this comment. If the author can not prepare the three replicates of RNAseq data, they must adequately explain the issues (robustness) to the reviewer, and must add the discussions of it to the discussion part in the next revision.

Response: We accept that at least three replications are required to conclude the differential expression level between two different tissues. However, this experiment was performed with the given replications. Therefore, we tried our best to overcome the limits of our experimental design: performing qRT-PCR with 10 upregulated genes to confirm the upregulation effect to avoid the false positive effects and comparing our expression data with Drosophila melanogaster expression database (Table sup. 3) to make sure the expression changes are sincere. Our sequencing result is in line with the qRT-PCR results and the expression datasets. Please gently reconsider our manuscript. Thank you so much for your effort and time for reviewing.

Added to Discussion: “In order to validate FPKM values of transcriptome data, ten genes were selected from the upregulated gene group, confirmed by qRT-PCR, and compared to transcriptome data (Figs. 3&4). Besides the qRT-PCR confirmation, to overcome the limit of experimental replications and to avoid false positive effects, the upregulated genes from the midgut of D. suzukii were compared with D. melanogaster’s expression database (Table S3)”